# The First Italian Telemedicine Program for Non-Critical COVID-19 Patients: Experience from Lodi (Italy)

**DOI:** 10.3390/jcm11185322

**Published:** 2022-09-10

**Authors:** Sara Forlani, Erminia Mastrosimone, Stefano Paglia, Simonetta Protti, Maria Paola Ferraris, Maria Chiara Casale, Mirko Di Capua, Maria Giuseppina Grossi, Marco Esposti, Davide Randazzo, Davide Giustivi

**Affiliations:** 1Telemitoring Center ASST Provincia di Lodi, 26900 Lodi, Italy; 2Data Management Unit ASST Provincia di Lodi, 26900 Lodi, Italy; 3A & E Department ASST Provincia di Lodi, 26900 Lodi, Italy

**Keywords:** COVID-19, telemedicine, emergency, e-health

## Abstract

Italy was the first Western country to face a massive SARS-CoV-2 outbreak. The limited information initially available on the natural course of the disease required caution in the discharge of patients accessing health facilities. This resulted in overcrowded health facilities and emergency services. In this context, improvements of other forms of hospital care assistance were needed. This study shows the results of the first Italian remote monitoring program for COVID-19 patients. The program was implemented by the Azienda Socio Sanitaria (ASST) Lodi (Italy) by using the innovative Zcare software^®^. Data generated during patient recruitment, monitoring, and discharge were extracted from the Zcare software and statistically analysed. Data refer to a sample of 1196 patients enrolled in the remote monitoring program in 2020. Patients reported symptoms mainly during the first week. The most frequently reported symptoms were general fatigue, cough, and loss of taste and smell (dysosmia). More than 80% of patients reported a saturation level below 96% at least once, and more than 70% had a temperature above 37 °C. Active monitoring of reported symptoms provided valuable insights into the disease’s natural history during its less severe acute phase. Only 109 individuals visited the emergency department at least once in the first 100 days of monitoring. Of these, 101 had COVID-19 infection, 69 of whom were hospitalized following a first clinical assessment at the emergency department. The ASST Lodi’s telemedicine strategy for COVID patients appears to be a viable alternative to hospitalization. This strategy enables the provision of proper care while making resources available for more critically ill patients, and enhances the availability of resources available for more critically ill patients.

## 1. Introduction

The COVID-19 pandemic radically changed people’s lives and habits around the world [1], leading to huge global burdens such as high rates of infection and death, financial difficulties, anxiety, and uncertainty-related stress [2]. Italy was one of the first countries to record a large number of illnesses. After the notification of the first case at the Codogno Emergency Department (Province of Lodi) in February 2020, the Italian National Institute of Health (ISS) recorded more than 101,500 cases of SARS-CoV-2 infection throughout the Italian nation by the end of April, and nearly 788,000 cases by the end of November, resulting in a peak infection rate of 1322 infected per 100,000 inhabitants [3].

One of the most difficult challenges faced by healthcare workers was the management of patients in overcrowded hospitals.

In that emergency situation, healthcare professionals looked for a possible solution in telemedicine, which, by leveraging technology, allowed them to remotely monitor patients, to improve selection criteria for hospitalization candidates and to provide the most suitable treatment for hospitalized patients [4]. According to the WHO definition, “telemedicine is the delivery of healthcare services, where distance is a critical factor, by all healthcare professionals using information and communication technologies for the exchange of valid information for the diagnosis, treatment, and prevention of disease and injuries, research and evaluation, and for continuing education of healthcare providers, all in the interest of advancing the health of individuals and their communities” [5].

Even though the National Guidelines for Telemedicine were developed in 2014 [6], in Italy remote monitoring strategies to manage COVID-19 patients were introduced much later. Despite years of delay in research and application, the need for technologically based services led to rapid advancements in telemedicine. In just 14 weeks, approximately 180 healthcare practices using digital solutions were introduced in Italy. Of these, 30% addressed COVID-19 patient treatment, and 70% fragile and chronic patients or cases under long-term treatment (cancer patients, heart patients, diabetics) [7]. The clinical procedure was initially carried out by using common tools already familiar to the patients, e.g., telephones and web platforms. Later, specific software for remote patient monitoring was developed.

This paper aims to report the findings when using ZCare software^®^ to monitor COVID-19 patients in the territory of Lodi (50 kilometres from Milan), one of the Italian areas most heavily affected by the SARS-CoV-2 virus in 2020. During this year, two epidemic waves with different characteristics were observed in Italy.

Data collected from remote monitoring of 1232 patients in 2020 were analysed with the following specific aims: to assess the main differences between patients monitored during the first and the second epidemic outbreaks, to determine how many patients enrolled in the telemedicine program who visited the emergency department (ED), to determine how many were admitted to the hospital after their conditions worsened and to evaluate benefits and opportunities of a implementing a telemedicine program.

## 2. Materials and Methods

This observational retrospective study makes use of the ZCare^®^ health telediagnosis software, specifically designed for home monitoring of COVID-19 patients and related data collection. The software tool was developed by Zucchetti S.p.A., a company based in Lodi, Italy.

The results are obtained by cross-referencing data collected by ZCare^®^ software during the monitoring phase with data of nasopharyngeal swabs and hospital admissions recorded in the Azienda Socio Sanitaria (ASST) Lodi.

Patients admitted to the telemonitoring program had a clinical picture indicative of a medium to high probability of SARS-CoV-2 infection, as well as mild to moderate symptoms. The definition of a suggestive clinical picture of SARS-CoV-2 infection was devised based on results of a rapid walk test. Specifically, patients were considered eligible after a negative rapid walk, validated as a Quick Walk Test (QWT), i.e., a walk of 30 to 40 m at the maximum possible speed for each patient [8]. Swab test positivity was not required for patient admission to remote care, because any negative results in patients showing clear core symptoms of COVID-19 were considered false negatives.

A total of 1232 patients were admitted to the telemonitoring program from 19 March to 31 December, but 36 patients abandoned the program on the first day. Out of these, 20 patients were later admitted to a hospital unit of the ASST Lodi.

In the case of 40 patients out of the remaining 1196, the available data were fragmented and difficult to analyse. Therefore, the study population consists of 1156 patients with a complete and analysable monitoring dataset (Figure 1).

As shown in Figure 2, the study population showed a well-balanced gender distribution, with 574 males (49.6%), and 582 females (50.3%). The largest group was aged 51–60 years (26% of the study population), with an average age of 56 years (SD = 16 years; mode = 54 years).

An informed consent was obtained by all patients admitted to remote monitoring. A health care professional gave each patient a pulse oximeter to daily assess blood oxygen saturation.

Patients were instructed to fill out an electronic questionnaire twice a day to report their main symptoms and a few clinical parameters (Table 1). Each patient uploaded their questionnaire into the ZCare^®^ platform twice a day, and those who were unable to transfer their data due to technological difficulties were assisted by a healthcare practitioner responsible for data collection and transfer to ZCare^®^.

Following data entry into the software, the healthcare worker had to evaluate results and to provide a score/color code based on the severity of patient’s conditions (Table 2)

White code: reported parameters and symptoms are normal, and no action is required by doctors and nurses.Green code: reported parameters and symptoms show a minimum deviation from normality; a phone check call from telemedicine nurses is required.Yellow code: moderate deviation from normality of vital parameters and symptoms; a phone check call from telemedicine nurses is urgent.Red code: vital parameters and symptoms are far from normal; it is urgent to consult a doctor and plan the patient’s transfer to the ED.

During the phone check call, the nurse evaluated patient status on the basis of symptoms and clinical features (e.g., fluency of speech, cough, oximetry, heart and respiratory rates). If the clinical conditions were deemed stable, a telephone test was administered to the patient, based on the scoring system described in Table 3. The patient was asked to count quickly from 1 to 20; as a second step the patient was asked to hold his breath for 20 s (normal conditions need the patient not to cough before 20 s in both steps). Eventually, it was necessary to measure blood oxygen saturation after walking 30 steps.

By combining the two scores described in Figure 3 and Figure 4, the following different scenarios can be identified (Figure 3):Green + Green or Yellow + Green: the telemedicine nurse must make a phone check call within 8 h.Yellow + Yellow: the patient’s parameters are not normal; a phone call from the telemedicine doctor is required. A home visit by the Unit for Continuity of Care and Integrated Home Assistance can be activated.Yellow + Red or Red + Red: the health conditions are getting worse, and the patient needs transfer to an ED. The doctor activates the transport of patient.

## 3. Results

Patients entered the telemonitoring program from March to December 2020, and were monitored for the time needed, depending on their clinical pathway. Remote monitoring was ended in case of recovery, transfer to the ED, hospitalization, or death. Consequently, different monitoring durations were observed. Two groups of patients have been analysed separately: patients monitored during the first wave of the epidemic from March to July 2020, and patients monitored during the second wave from August to December 2020. Characteristics of patients in the two epidemic waves are presented in Table 4. A greater number of patients were monitored during the first wave. Moreover, the two groups differ with respect to the duration of monitoring, which was shorter during the second wave.

### 3.1. Analysis of Reported Symptoms

In general, for all symptoms, the number of reports decreased over time (Figure 4), with the highest numbers observed during the first five days of monitoring. From day 15 on, a 67% drop was observed. Fatigue, cough, lack of appetite, and muscle pain were the most frequently patient-reported symptoms. Dysosmia recovered more slowly than the other symptoms (yellow curve).

By comparing data collected during the first and the second wave, differences were seen in the percentage of patients reporting a given symptom. These differences are shown in Figure 5, for each symptom.

In general, a higher percentage of patient reports were observed during the second wave as compared to the first wave. One of the most relevant differences concerns the lower percentage of respiratory distress reports during the second wave (−7%). The greatest percentage difference relates to the symptom dysosmia, with more patients reporting it during the second wave (+18%).

### 3.2. Analysis of Vital Signs

Blood oxygenation was the first vital sign examined. SARS-CoV-2 causes respiratory symptoms including a substantial pulmonary dysfunction, with worsening arterial hypoxemia, eventually leading to acute respiratory distress syndrome. Regular monitoring of oxygen saturation levels can help identify a more serious pulmonary disease.

Of 780 patients monitored during the first wave, a total of 698 reported oxygen saturation level below 96% (indicating hypoxemia) during the first week, 504 patients during the second week, and 365 patients in the third week (Figure 6). Reports of this symptom were significantly reduced from the fourth week on.

The trend of reports of hypoxemia did not vary between the two waves. Patients reporting an oxygen saturation level below 96% were concentrated in the first two weeks, and a noticeable decrease was observed following the third week.

Fever is one of the most prevalent symptoms of COVID-19. All cases reporting a body temperature above 37 °C, suggestive of fever, were investigated.

Considering the whole sample of 1156 monitored cases, 810 patients (70%) reported fever with an average recorded temperature of 37.5 °C and a modal value of 38 °C. The highest recorded temperature was 40 °C, reported by four patients.

The trend of fever reports did not vary between the first and the second wave (Figure 7). The highest concentration of reports of fever was observed during the first days, whereas a relatively small number of patients reported fever in the end-monitoring phase.

### 3.3. Monitoring Patient-Doctor Interactions

Information about patient–doctor interactions was obtained from the operator’s clinical diaries completed during the telephone conversation with the patient.

As seen in Figure 8, 948 patients needed a phone call by a nurse, accounting for 82% of all cases participating in the telemonitoring program. Only 208 patients showed a stable white code, so they were called only for discharge contact. All contacted patients spoke to a nurse at least once, and 755 patients were also contacted by a doctor, for a more detailed consultation and health assessment. In the case of 85 patients, oxygen therapy had to be remotely activated by a doctor, after assessment of vital signs, quality of speech, and patient-reported dyspnoea.

Hundred days after the beginning of the surveillance program, data on ED access and hospital admissions were analysed. The hospitalization and ED access observation period runs from February 2020 to February 2021. An ED visit occurring after more than 100 days of observation was considered not linked to the infection recorded when the patient was enrolled in the program.

A total of 94 patients accessed the ED of the ASST Lodi at least once after the start of the telemonitoring program, 25 of them on doctor’s recommendation, and 69 of them on a voluntary base or following nurse advice. None of the patients who were prescribed home oxygen therapy required admission to the ED.

As shown in Figure 9, out of the 94 patients accessing the ED, 39 cases were discharged, and 55 patients were hospitalized. Of these, four died due to severe COVID-19 and another four cases had to be admitted to the intensive care unit (ICU) for a critical health condition.

## 4. Discussion

Many healthcare professionals look at telemedicine as to the future of medical practice, because it includes many features such as:teleeducation, which allows doctors to stay up to date on new clinical practices while reducing inactivity time and encouraging dialogue between academic and community doctors;teleconsultation, which allows clinicians to easily obtain, record, and compare patient data and clinical information useful in the diagnosis and treatment process.remote monitoring of chronic patients’ conditions, which facilitates the management of chronic health and aids the monitoring of the patients’ compliance with prescriptions [9];remote monitoring of acute isolated patients’ parameters and conditions, which ensures adequate safety of healthcare professionals, restraining direct contact with the patient only when strictly necessary, and allowing patients with suspected infection-related symptoms and positive asymptomatic subjects to be followed with regular checks while maintaining social distancing and preventing the overcrowding of EDs and general practitioner offices;screening, as recent studies have demonstrated that using telemedicine techniques to conduct screenings might be helpful to prevent the worsening of diseases and chronic conditions [10];promotion of continuity of care, by allowing the outpatient management of periodic visits otherwise interrupted due to extraordinary events, such as an epidemic or a pandemic, according to the dispositions of the local administrations [11].

Since the first outbreak of COVID-19 in Italy, social distancing has proven to be a key measure to limit the spreading of viral infection caused by SARS-CoV-2. The Lodi ASST telemonitoring program, has proven to be an effective model of care for treating those with less serious infections.

Monitoring was carried out by a team of nurses and doctors who were experts in remote supervision and management of cardiac and respiratory diseases, following the latest guidelines available on home management of mild COVID-19 cases [12].

Considering patients who completed the telemonitoring program, the high extent of participation was a significant measure of success, given the relatively small number of abandonments recorded. All patients, who had been instructed during the recruitment phase on how to detect their vital parameters, continued to upload signs and symptoms until the last day of monitoring.

The analysis of the medical contacts show that patients were never abandoned during monitoring; more than 80% of patients received a phone call from a nurse or a doctor to check their health status. These continuous contacts with a healthcare professional have allowed to limit inappropriate accesses to the ED and to ensure the patient has an appropriate therapy, even if affected by more severe respiratory conditions. Moreover, participation in the telemonitoring program gave the patient a sense of trust and assuredness, thus facilitating a good compliance with the monitoring program until discharge.

The results show no relevant change in symptomatology between the two samples monitored during the first and second epidemic wave. In both cases patients suffered generic symptoms including fatigue, muscle pain, headaches, as well as respiratory symptoms like coughing and breathing distress.

Compared to the other symptoms, dysosmia showed the slowest recovery rate and was the most persistent. Typically, patients reported symptoms during the first week of monitoring, with a peak of complaints occurring between the second and the third day.

The low number of patients who accessed the ED or were hospitalized during the first 100 days after enrollment in the telemonitoring program is an indicator of success of the remote management strategy. The availability of clinical data and digital recordings for each event enables effective patient management, time, and resource savings.

Telemedicine significantly reduces the time needed for clinical assessments, allowing healthcare professionals to safely take care of more patients, avoid hospital and clinic overcrowding, and outreach to those who have limited access to care for various reasons such as mobility issues or rural habitations. It also provides for faster contact between professionals, allowing for a multidisciplinary approach to patient treatment and access to high-quality healthcare facilities and highly individualized therapeutic services. The possibility of establishing a connection between doctor and patient physically separated led to reduction in emergency room access and hospital admissions, as well as a stronger adherence of patients to the prescribed treatments, resulting in fewer hospital readmissions. If accessibility and convenience are the most important factors in patient-centred care, telemedicine can improve the efficiency and organization of healthcare [13].

Telemedicine has proven to be the most cost-effective alternative to inpatient care for COVID-19 patients. It is considered a safe service solution, useful for maintaining social distancing and reducing virus circulation, and is effective in reducing hospitalization and access to emergency settings. To implement a technology-based assistance model, healthcare professionals must be adequately trained. Furthermore, it is crucial to fulfil the needs of patients through proper education and coaching during remote monitoring, and to adopt a user-friendly tool, to make the new clinical practice easier and more acceptable by both patients and health professionals. The activation of the telemonitoring program based on the ZCare^®^ software in the management of chronic patients with monitoring their adherence to prescriptions, could be an interesting and feasible future direction at ASST Lodi.

### Study Limitations

This is a retrospective single-centre study that would need to be extended to other centres, but, due to the emergency context in which the whole telemonitoring protocol was designed, this study is, hopefully, hardly repeatable.

In literature we have not found any other telemonitoring project comparable to ours, but many different experiences prove the safety and effectiveness of telemedicine and suggest its implementation at scale in healthcare systems [14].

Despite these limitations, we consider the results of this study valid as they show how, in an area as small as the Province of Lodi, telemonitoring nonsevere patients affected by a little-known illness in an epidemic context has proven to be valid support for avoiding overcrowded EDs and conducting the safe remote monitoring of patients.

## Figures and Tables

**Figure 1 jcm-11-05322-f001:**
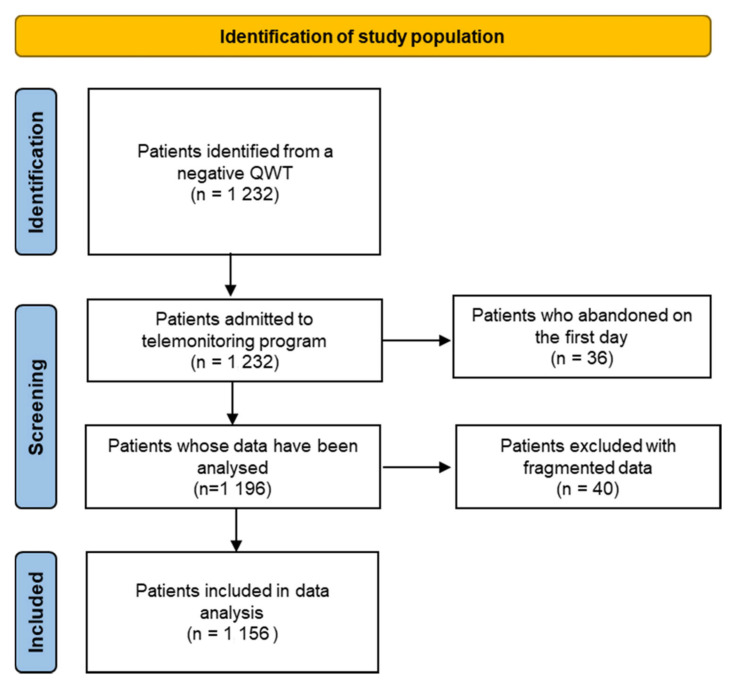
Identification of study population.

**Figure 2 jcm-11-05322-f002:**
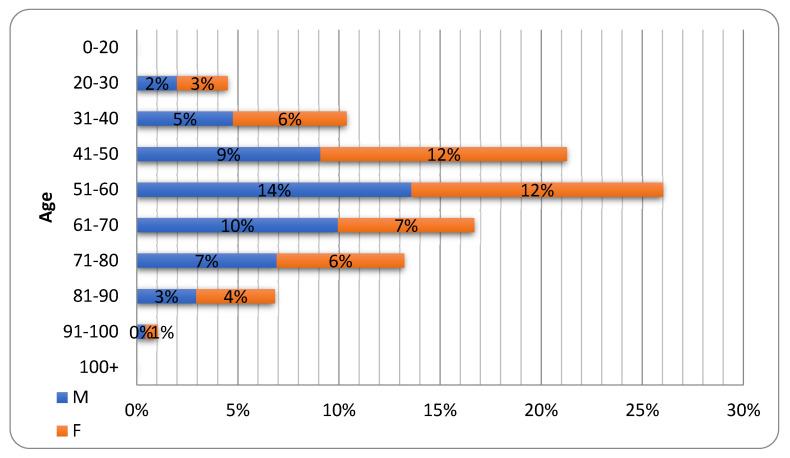
Patient distribution by age and gender.

**Figure 3 jcm-11-05322-f003:**
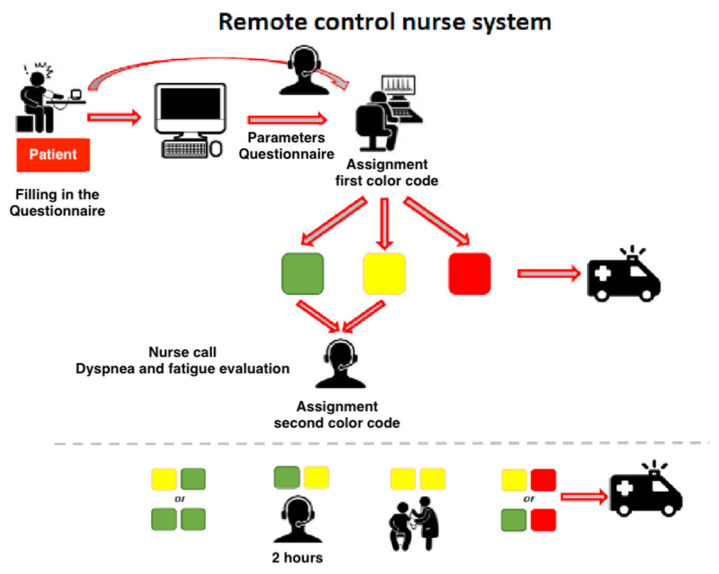
Remote control nurse system.

**Figure 4 jcm-11-05322-f004:**
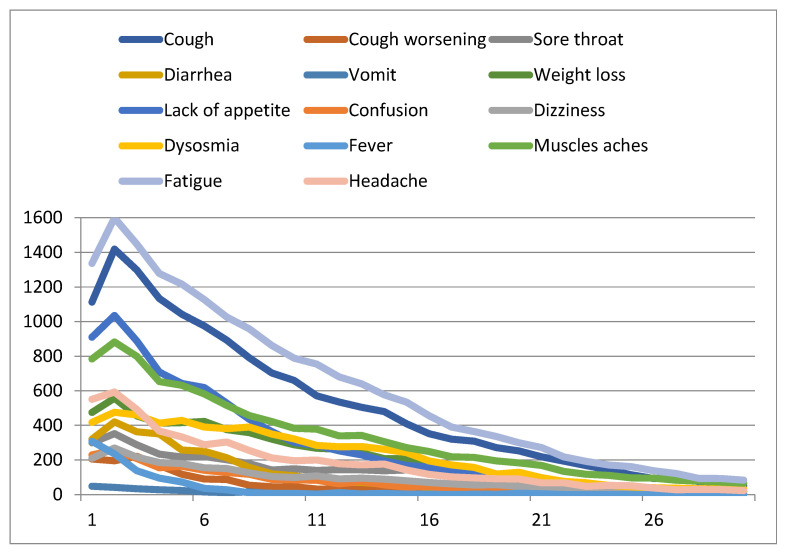
Number of reported symptoms per monitoring day.

**Figure 5 jcm-11-05322-f005:**
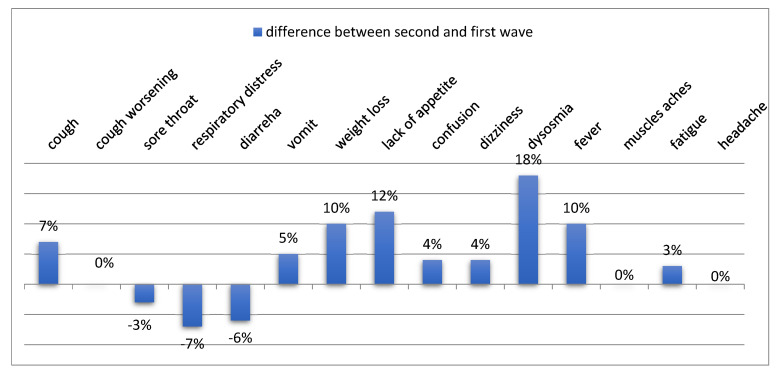
Percentage of patients reporting each symptom: differences between the two waves.

**Figure 6 jcm-11-05322-f006:**
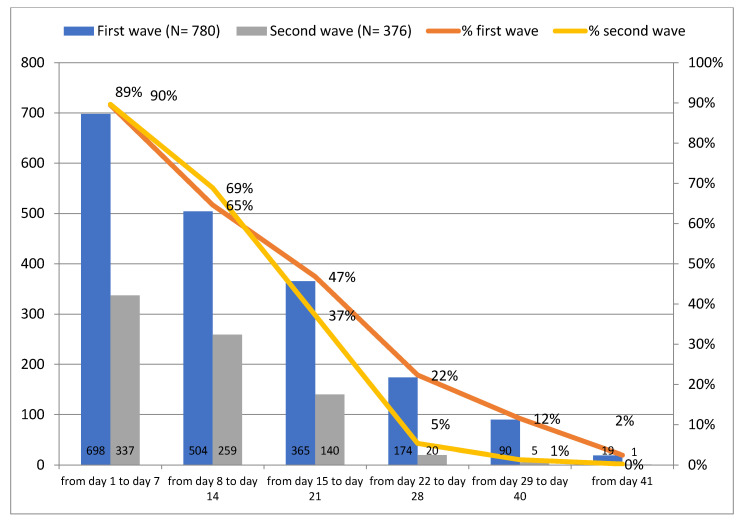
Reports of SpO2 < 96% during the first and second waves.

**Figure 7 jcm-11-05322-f007:**
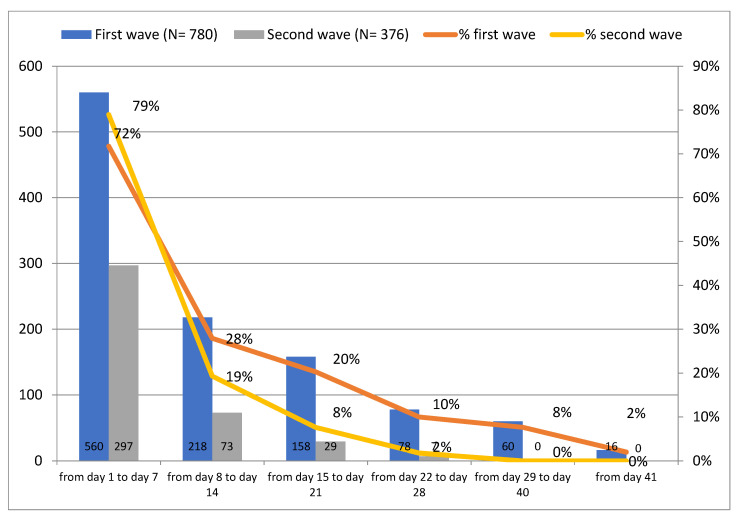
Patients with T > 37 °C per monitoring day.

**Figure 8 jcm-11-05322-f008:**
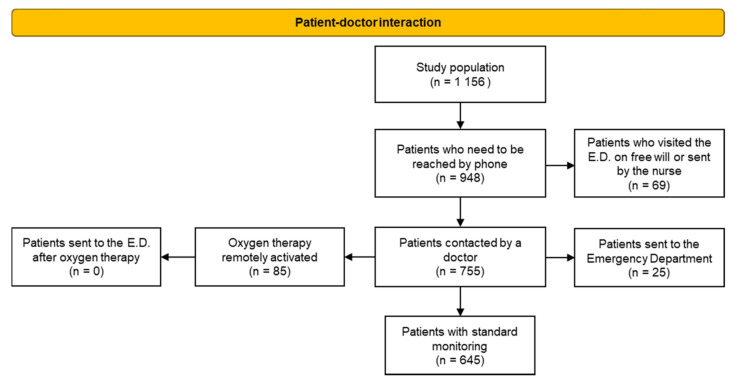
Patient-doctor interaction.

**Figure 9 jcm-11-05322-f009:**
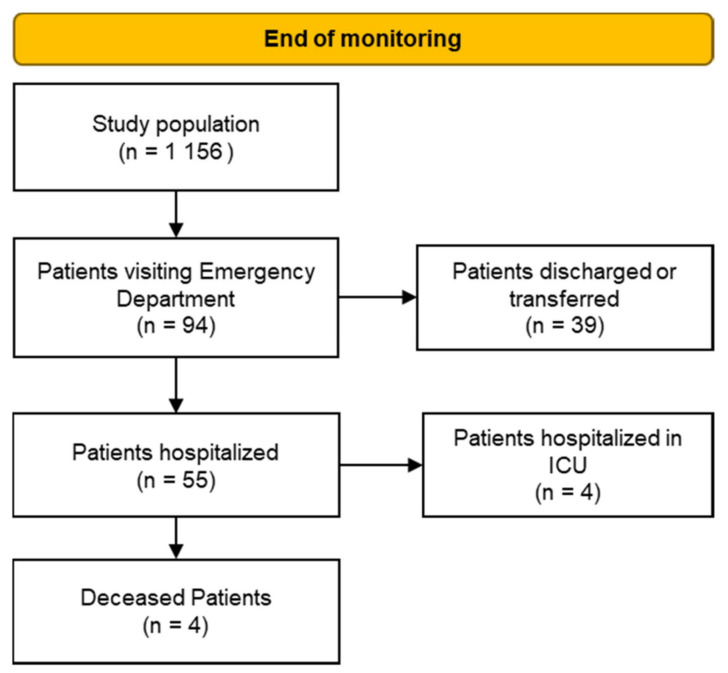
Study patients who accessed the ED.

**Table 1 jcm-11-05322-t001:** Monitored symptoms and clinical parameters.

Section	Symptoms/Parameters
Respiratory	Cough Cough worsening
	Sore throat
	Respiratory Distress
Gastroenteric	Diarrhoea
	Vomit
	Weight loss Lack of appetite
Nervous	Confusion
	Dizziness
Sense	Dysosmia
Generic	Fever
	Headache
	Fatigue
	Muscle aches
Clinical Parameters	Oxygen saturation
	Heart rate Respiratory rate Temperature Systolic pressure

**Table 2 jcm-11-05322-t002:** Score parameters and colour code.

Score Parameters							
Parameters	3	2	1	0	1	2	3
**RR**	≤8		9–11	12–20		21–24	≥25
**SpO2**	≤91	92–93	94–95	≥96			
**Temp.**	≤35		35.1–36	36–38	38.1–39	≥39.1	
**PAS**	≤90	91–100	101–110	111–219			≥220
**HR**	≤40		41–50	51–90	91–110	111–130	≥131

RR, respiratory rate; SpO2, blood oxygen saturation; Temp., body temperature; PAS, systolic blood pressure; HR, heart rate.

**Table 3 jcm-11-05322-t003:** Nurse assessment colour code.

Parameters	Code		
Shiver	YES		NO
Onset of pause during speech (s)	<5	5–10	>10
Hold apnea (s)	<10		>10
SpO2 after 30 steps (%)	<90	90–95	>95

**Table 4 jcm-11-05322-t004:** Characteristics of patients by epidemic waves.

Characteristics of Patients by Epidemic Waves	First Wave March 2020–July 2020	Second Wave August 2020–December 2020
N. of monitored patients	780	376
Sex	374 Male	203 Male
	406 Female	173 Female
Average age	56 years	56 years
Average monitoring duration	20 days	15 days

## Data Availability

Not applicable.

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
