# Peer review of "The First Italian Telemedicine Program for Non-Critical COVID-19 Patients: Experience from Lodi (Italy)"

_jcm, 2022, doi:10.3390/jcm11185322_

Round 1
Reviewer 1 Report
I have read with great interest this manuscript which emphasizes the usefulness of telemedicine in the management of non-critical COVID-19 patients.
However, this reviewer raises some issues that should be addressed by the authors.
1- The manuscript lacks a study limitation section that authors should add at the end of the discussion.
2- The discussion is quite short. Moreover, the bibliography is extremely scarce (only 8 references). The importance of the use of telemedicine, especially during the COVID-19 pandemic, has been underlined by some recent manuscripts that should be added as well as commented on in the discussion (1- J Diabetes Res. 2020 Oct 14; 2020:9036847. doi: 10.1155/2020/9036847. 2- Diabetes Metab Res Rev. 2019 Mar; 35(3):e3113. doi: 10.1002/dmrr.3113.)
3- A linguistic revision by a native English speaker is required.
Author Response
Dear Colleague,
Thank you for your kind words and your punctual revision.
We read with much interest the articles you suggested and tried to expand the discussion section and the bibliography; moreover we had the manuscript checked by an English native speaker colleague.
We uploaded the new manuscript and wait for your precious revision.
Thank you again
Sara Forlani
Reviewer 2 Report
Dear Colleagues!
Congratulations on such an excellent and novel telemedicine strategy and teamwork, exemplified by doctors, nurses, and health officials, that I feel is an outstanding approach to relieve emergency rooms and realistically improve patient care in general.
I have a few small corrections to make such as.
Line 126: HR: heart rate is misspelled
Fig 5: Under Remote control nurse system; should read "parameters"
Line 231: Perhaps for clarity, instead of "had access to the Emergency Department" it should read "did not require admission to the Emergency Department".
Author Response
Dear Colleague,
Thank you for your kind words, your review and for your appreciation of our work
We proceeded to apply the corrections you made and to have our manuscript checked by an English native speaker colleague.
We submit the new draft for your kind new review
Thank you again
Sara Forlani
Round 2
Reviewer 1 Report
No further comments.